# Why Does Decentralized Training Outperform Synchronous Training in the Large Batch Setting?

## Abstract

Distributed Deep Learning (DDL) is essential for large-scale Deep Learning (DL) training. Using a sufficiently large batch size is critical to achieving DDL runtime speedup. In a large batch setting, the learning rate must be increased to compensate for the reduced number of parameter updates. However, a large batch size may converge to sharp minima with poor generalization, and a large learning rate may harm convergence. Synchronous Stochastic Gradient Descent (SSGD) is the de facto DDL optimization method. Recently, Decentralized Parallel SGD (DPSGD) has been proven to achieve a similar convergence rate as SGD and to guarantee linear speedup for non-convex optimization problems. While there was anecdotal evidence that DPSGD outperforms SSGD in the large-batch setting, no systematic study has been conducted to explain why this is the case. Based on a detailed analysis of the DPSGD learning dynamics, we find that DPSGD introduces additional landscape-dependent noise, which has two benefits in the large-batch setting: 1) it automatically adjusts the learning rate to improve convergence; 2) it enhances weight space search by escaping local traps (e.g., saddle points) to find flat minima with better generalization. We conduct extensive studies over 12 state-of-the-art DL models/tasks and demonstrate that DPSGD consistently outperforms SSGD in the large batch setting; and DPSGD converges in cases where SSGD diverges for large learning rates. Our findings are consistent across different application domains, Computer Vision and Automatic Speech Recognition, and different neural network models, Convolutional Neural Networks and Long Short-Term Memory Recurrent Neural Networks.

## 1 Introduction

Deep Learning (DL) has revolutionized AI training across application domains: Computer Vision (CV) (Krizhevsky et al., 2012; He et al., 2015), Natural Language Processing (NLP) (Vaswani et al., 2017), and Automatic Speech Recognition (ASR) (Hinton et al., 2012). Stochastic Gradient Descent (SGD) is the fundamental optimization method used in DL training. Due to massive computational requirements, Distributed Deep Learning (DDL) is the preferred mechanism to train large scale Deep Learning (DL) tasks. In the early days, Parameter Server (PS) based Asynchronous SGD (ASGD) training was the preferred DDL approach (Dean et al., 2012; Li et al, 2014) as it did not require strict system-wide synchronization. Recently, ASGD has lost popularity due to its unpredictability and often inferior convergence behavior (Zhang et al., 2016b). Practitioners now favor deploying Synchronous SGD (SSGD) on homogeneous High Performance Computing (HPC) systems. The degree of parallelism in a DDL system is dictated by batch size: the larger the batch size, the more parallelism and higher speedup can be expected. However, large batches require a larger learning rate and overall they may negatively affect model accuracy because 1) large batch training usually converges to sharp minima which do not generalize well (Keskar et al., 2016) and 2) large learning rates may violate the conditions (i.e., the smoothness parameter) required for convergence in nonconvex optimization theory (Ghadimi & Lan, 2013). Although training longer with large batches could lead to better generalization (Hoffer et al., 2017), doing so gives up some or all of the speedup we seek. Through meticulous hyper-parameter design (e.g., learning rate) tailored to each specific task, SSGD-based DDL systems have enabled large batch training and shortened training time for some challenging CV tasks (Goyal et al., 2017; You et al., 2017) and NLP tasks (You et al., 2019) from weeks to hours or less. However, it is observed that SSGD with large batch size leads to large training loss and inferior model quality for ASR tasks (Zhang et al., 2019b), as illustrated in Figure 1a (red curve). In this paper we found for other types of tasks (e.g. CV) and DL models, large batch SSGD has the same problem (Figure 1b and Figure 1c). The cause of this problem could be that training gets trapped at saddle points since large batches reduce the magnitude of noise in the

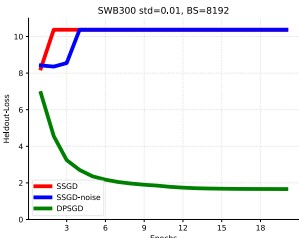 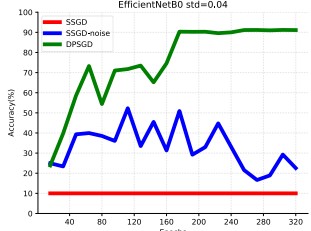 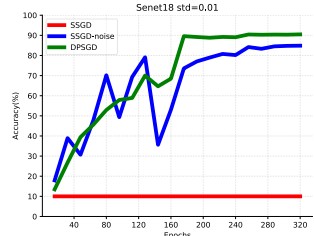

(a) LSTM, SWB300, BS 8192   (b) EfficientNet, CIFAR-10, BS 8192   (c) SENet-18, CIFAR-10, BS 8192

Figure 1: SSGD (red) does not converge in the large batch setting. Figure 1a plots the heldout-loss, the lower the better. Figure 1b and Figure 1c plot the model accuracy, the higher the better. By injecting Gaussian noise, SSGD might escape early traps but result in much worse model (blue) compared to DPSGD (green) in the large batch setting. The detailed task descriptions and training recipes are described in Section 4.3. BS stands for Batch-Size.

stochastic gradient and prevent the algorithm from exploring the whole parameter space. To solve this problem, one may add isotropic noise (e.g., spherical Gaussian) to help SSGD escape from saddle points (Ge et al., 2015). However, this is not a good solution for high-dimensional DL training as shown in the blue curves of Figure 1. One possible reason is that the complexity of escaping a saddle point by adding isotropic noise has a polynomial dependency on the dimension of the parameter space, so adding such noise in a high dimensional space (such as deep learning) does not bring significant benefits. In this paper, we have found that Decentralized Parallel SGD (DPSGD) (Lian et al., 2017b) greatly improves large batch training performance, as illustrated in the green curves in Figure 1. Unlike SSGD, where each learner updates its weights by taking a global average of all learners' weights, DPSGD updates each learner's weights by taking a partial average (i.e., across a subset of neighboring learners). Therefore, in DPSGD, each learner's weights differ from the weights of other learners.[1] The key difference among SSGD, SSGD with Gaussian noise [2] and DPSGD is the source of noise during the update, and this noise directly affects performance in deep learning. This naturally motivates us to study *Why decentralized training outperform synchronous training in the large batch setting?* More specifically, we try to understand whether their performance difference is caused by their different noise. We answer these questions from both theoretical and empirical perspectives. Our contributions are:

- We analyze the dynamics of DDL algorithms, including both SSGD and DPSGD. We show, both theoretically and empirically, that the *intrinsic noise* in DPSGD can 1) reduce the effective learning rate when the gradient is large to help convergence; 2) enhance the search in weight space for flat minima with better generalization.
- We conduct extensive empirical studies of 12 CV and ASR tasks with state-of-the-art CNN and LSTM models. Our experimental results demonstrate that DPSGD consistently outperforms SSGD, across application domains and Neural Network (NN) architectures in the large batch setting, *without any hyper-parameter tuning*. To the best of our knowledge, we are unaware of any generic algorithm that can improve SSGD large batch training on this many models/tasks.

The remainder of this paper is organized as follows. Section 2 details the problem formulation and learning dynamics analysis of SSGD, SSGD+Gaussian, and DPSGD; Section 3 and Section 4 detail the empirical results; and Section 5 concludes the paper.

## 2   ANALYSIS OF STOCHASTIC LEARNING DYNAMICS AND EFFECTS OF LANDSCAPE-DEPENDENT NOISE

We first formulate the dynamics of an SGD based learning algorithm with multiple ($n > 1$) learners indexed by $j = 1, 2, 3, ...n$ following the same theoretical framework established for a single learner (Chaudhari & Soatto, 2018). At each given time (iteration) $t$, each learner has its own weight vector $\vec{w}_j(t)$, and the average weight vector $\vec{w}_a(t)$ is defined as: $\vec{w}_a(t) \equiv n^{-1} \sum_{j=1}^{n} \vec{w}_j(t)$.

---

[1] The detailed DPSGD algorithm and its learning dynamics are described in Section 2.

[2] We use the terms "SSGD with Gaussian noise" and "SSGD*" interchangeably in this paper.

Each learner $j$ updates its weight vector according to the cross-entropy loss function $L^{\mu_j(t)}(\vec{w})$ for minibatch $\mu_j(t)$ that is assigned to it at time $t$. The size of the local minibatch is $B$, and the overall batch size for all learners is $nB$. Two multi-learner algorithms are described below.

**(1) Synchronous Stochastic Gradient Descent (SSGD):** In the synchronous algorithm, the learner $j \in [1, n]$ starts from the average weight vector $\vec{w}_a$ and moves along the gradient of its local loss function $L^{\mu_j(t)}$ evaluated at the average weight $\vec{w}_a$:

$$\vec{w}_j(t+1) = \vec{w}_a(t) - \alpha \nabla L^{\mu_j(t)}(\vec{w}_a(t)), \tag{1}$$

where $\alpha$ is the learning rate.

**(2) Decentralized Parallel SGD (DPSGD):** In the DPSGD algorithm (Lian et al., 2017a), learner $j$ computes the gradient at its own local weight $\vec{w}_j(t)$. The learning dynamics follows:

$$\vec{w}_j(t+1) = \vec{w}_{s,j}(t) - \alpha \nabla L^{\mu_j(t)}(\vec{w}_j(t)). \tag{2}$$

where $\vec{w}_{s,j}(t)$ is the starting weight set to be the average weight of a subset of "neighboring" learners of learner-$j$, which corresponds to the non-zero entries in the mixing matrix defined in (Lian et al., 2017a) (note that $\vec{w}_{s,j} = \vec{w}_a$ if all learners are included as neighbors).

By averaging over all learners, the learning dynamics for the average weight $\vec{w}_a$ for both SSGD and DPSGD can be written formally the same way as: $\vec{w}_a(t+1) = \vec{w}_a(t) - \alpha \vec{g}_a$, where $\vec{g}_a = n^{-1} \sum_{j=1}^n \vec{g}_j$ is the average gradient and $\vec{g}_j$ is the gradient from learner-$j$. The difference between SSGD and DPSGD is the weight at which $\vec{g}_j$ is computed: $\vec{g}_j \equiv \nabla L^{\mu_j(t)}(\vec{w}_a(t))$ is computed at $\vec{w}_a$ for SSGD; $\vec{g}_j \equiv \nabla L^{\mu_j(t)}(\vec{w}_j(t))$ is computed at $\vec{w}_j$ for DPSGD.

By projecting the weight displacement vector $\Delta \vec{w}_a \equiv \alpha \vec{g}_a$ onto the direction of the gradient $\vec{g} \equiv \nabla L(\vec{w}_a)$ of the overall loss function $L$ at $\vec{w}_a$, we can write the learning dynamics as:

$$\vec{w}_a(t+1) = \vec{w}_a(t) - \alpha_e \vec{g} + \vec{\eta}, \tag{3}$$

where $\alpha_e \equiv \alpha \vec{g}_a \cdot \vec{g}/||\vec{g}||^2$ is an effective learning rate and $\vec{\eta} = -\alpha \vec{g}_a + \alpha_e \vec{g}$ is the noise term that describes the stochastic weight dynamics in directions orthogonal to $\vec{g}$. The noise term has zero mean $\langle \vec{\eta} \rangle_\mu = 0$ and its strength is characterized by its variance $\Delta(t) \equiv ||\vec{\eta}||^2$. $\Delta$ and $\alpha_e$ are related by the equality: $\alpha_e^2 ||\vec{g}||^2 + \Delta = \alpha^2 ||\vec{g}_a||^2$, which indicates that a higher noise strength $\Delta$ leads to a lower effective learning rate $\alpha_e$.

The noise strength $\Delta$ (and hence $\alpha_e$) is different in SSGD and DPSGD. The DPSGD noise $\Delta_{DP}$ is larger than the SSGD noise $\Delta_S$ by an additional noise $\Delta^{(2)}(>0)$ that originates from the difference of local weights ($\vec{w}_j$) from their mean ($\vec{w}_a$): $\Delta_{DP} = \Delta_S + \Delta^{(2)}$, see Appendix B for details. By expanding $\Delta^{(2)}$ w.r.t. $\Delta \vec{w}_j \equiv \vec{w}_j - \vec{w}_a$, we obtain the average $\Delta^{(2)}$ over minibatch ensemble $\{\mu\}$:

$$\langle \Delta^{(2)} \rangle_\mu \equiv \alpha^2 \langle ||n^{-1} \sum_{j=1}^n [\nabla L^{\mu_j}(\vec{w}_j) - \nabla L^{\mu_j}(\vec{w}_a)]||^2 \rangle_\mu \approx \alpha^2 \sum_{k,l,l'} H_{kl} H_{kl'} C_{ll'}, \tag{4}$$

where $H_{kl} = \nabla_{kl}^2 L$ is the Hessian matrix of the loss function and $C_{ll'} = n^{-2} \sum_{j=1}^n \Delta w_{j,l} \Delta w_{j,l'}$ is the weight covariance matrix. It is clear that $\Delta^{(2)}$ depends on the loss landscape – it is larger in rough landscapes and smaller in flat landscapes.

It is important to stress that the noise $\vec{\eta}$ in Eq.3 is not an artificially added noise. It is intrinsic to the use of minibatches (random subsampling) in SGD-based algorithms and is enhanced by the difference among different learners in DPSGD. The noise strength $\Delta$ varies in weight space via its dependence on the loss landscape, as explicitly shown in Eq.4. However, besides its landscape dependence, SGD noise also depends inversely on the minibatch size $B$ (Chaudhari & Soatto, 2018). With $n$ synchronized learners, the noise in SSGD scales as $1/(nB)$, which is too small to be effective for a large batch size $nB$. A main finding of our paper is that the additional landscape-dependent noise $\Delta^{(2)}$ in DPSGD can make up for the small SSGD noise when $nB$ is large and help enhance convergence and generalization in the large batch setting.

In the following, we investigate the effects of this landscape-dependent noise for SSGD and DPSGD using the MNIST dataset where each learner is a fully connected network with two hidden layers (50 units per layer). We focus on the large batch setting using $nB = 2000$ in the experiments.

## 2.1 Noise in DPSGD reduces effective learning rate to help convergence

First, we study a case with a large learning rate $\alpha = 1$. In this experiment, we used $n = 5$, and $\vec{w}_{s,j} = \vec{w}_a$ for DPSGD. As shown in the upper panel of Fig. 2(a), DPSGD converges to a solution with low loss (2.1% test error), but SSGD fails to converge. As shown in Fig. 2(a) (lower panel), the effective learning rate $\alpha_e$ is reduced in DPSGD during early training ($0 \leq t \leq 700$) while $\alpha_e$ in SSGD remains roughly the same as $\alpha$. This reduction of $\alpha_e$ caused by the stronger noise $\Delta$ in DPSGD is essential for convergence by avoiding overshoots when gradients are large in the beginning of the training process. In the later stage of the training process when gradients are smaller, the landscape-dependent DPSGD noise decreases and $\alpha_e$ increases back to be $\approx \alpha$. To show the importance of the landscape-dependent noise, we introduce a variant of SSGD, SSGD*, by injecting a Gaussian noise with a constant variance to weights in SSGD. However, most choices of this injected noise fail to converge. Only by fine tuning the injected noise strength can SSGD* converge, but to an inferior solution with much higher loss and test error (5.7%). The poor performance is likely due to the persistent reduction of $\alpha_e$ even in the later stage of training (see Fig. 2(a) (lower panel)) since the added Gaussian noise in SSGD* is independent of the loss landscape.

This insight on reducing learning rate is consistent with nonconvex optimization theory (Ghadimi & Lan, 2013; Lian et al., 2017b). When we use a larger batch size, stochastic gradient has smaller variance, and nonconvex optimization is able to choose a larger learning rate without affecting its convergence. However, the learning rate should be limited by $1/l_s$ where $l_s$ is the smoothness parameter. In the very large batch setting, the learning rate under the linear scaling rule (Goyal et al., 2017) may indeed exceed this limit ($1/l_s$). Here, we show that these conflicting requirements can be resolved in DPSGD where the enhanced landscape-dependent noise adaptively adjusts the effective learning rate by reducing $\alpha_e$ when the loss landscape is rough with large gradients and restoring to the original large $\alpha$ when the landscape is smooth. In Appendix E, we consider a simple synthetic problem where we show that the larger noise in DPSGD allows the algorithm to escape saddle points in the loss function landscape while the SSGD algorithm gets stuck for a long time.

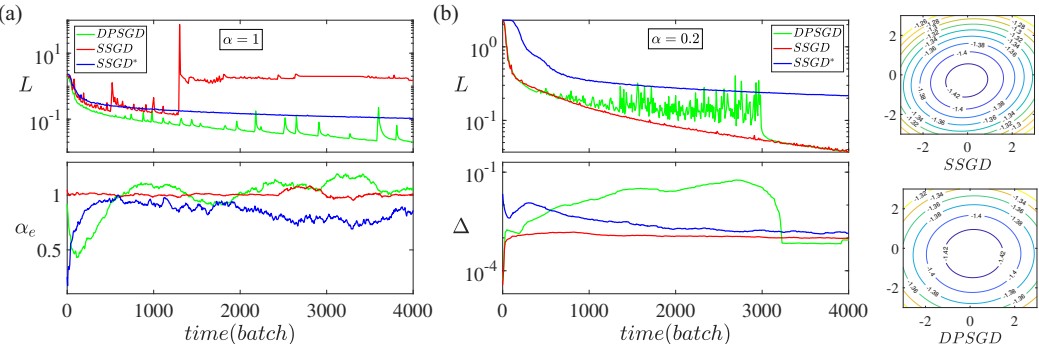

Figure 2: Comparison of different multi-learner algorithms, DPSGD (green), SSGD (red), and SSGD* (blue). (a) For a large learning rate $\alpha = 1$, the lowered effective learning rate $\alpha_e$ in DPSGD in the beginning of training allows DPSGD to converge while SSGD fails to converge. SSGD* also converges but to an inferior solution. (b) For a smaller $\alpha = 0.2$, DPSGD finds a flatter minimum with a lower test error than SSGD. SSGD* has the worst performance. See text for detailed description.

## 2.2 Noise in DPSGD enhances search to find flat minima with better generalization

Next, we consider a case with a smaller learning rate $\alpha = 0.2$. Here we used $n = 6$ and $\vec{w}_{s,j}(t)$ in DPSGD is the average weight of 2 neighbors on each side. In this case, both SSGD and DPSGD can converge to a solution, but their learning dynamics are different. As shown in Fig. 2(b) (upper panel), while the training loss $L$ of SSGD (red) decreases smoothly, the DPSGD training loss (green) fluctuates widely during the time window (1000-3000) when it stays significantly above the SSGD training loss. As shown in Fig. 2(b) (lower panel), these large fluctuations in $L$ are caused by the high and increasing noise level in DPSGD. This elevated noise level in DPSGD allows the algorithm to search in a wider region in weight space. At around time 3000(batch), the DPSGD loss decreases suddenly and eventually converges to a solution with a similar training loss as SSGD. However, despite their similar final training loss, the DPSGD loss landscape is flatter (contour lines further

| | CIFAR10 | | | | | |
|---|---|---|---|---|---|---|
| | EfficientNet-B0 | SENet-18 | VGG-19 | ResNet-18 | DenseNet-121 | MobileNet |
| Size | 11.11 MB | 42.95 MB | 76.45 MB | 42.63 MB | 26.54 MB | 12.27 MB |
| Time | 2.92 Hr | 1.58 Hr | 1.08 Hr | 1.37 Hr | 5.48 Hr | 1.02 Hr |
| | CIFAR10 | | | | SWB300 | SWB2000 |
| | MobileNetV2 | ShuffleNet | GoogleNet | ResNext-29 | LSTM | LSTM |
| Size | 8.76 MB | 4.82 MB | 23.53 MB | 34.82 MB | 164.62 MB | 164.62 MB |
| Time | 1.96 Hr | 2.46 Hr | 5.31 Hr | 4.55 Hr | 26.88 Hr | 203.21 Hr |

Table 1: Evaluated workload model size and training time. Training time is measured when running on 1 V100 GPU. CIFAR-10 is trained with batch size 128 for 320 epochs. SWB-300 and SWB-2000 are trained with batch size 128 for 16 epochs.

apart) than SSGD landscape. Remarkably, the DPSGD solution has a lower test error (2.3%) than the test error of the SSGD solution (2.6%). We have also tried the SSGD* algorithm, but the performance (3.9% test error) is worse than both $SSGD$ and $DPSGD$.

To understand their different generalization performance, we studied the loss function landscape around the SSGD and DPSGD solutions. The contour plots of the loss function $L$ around the two solutions are shown in the two right panels in Fig. 2(b). We found that the loss landscape near the DPSGD solution is flatter than the landscape near the SSGD solution despite having the same minimum loss. Our observation is consistent with (Keskar et al., 2016) where it was found that SSGD with a large batch size converges to a sharp minimum which does not generalize well. Our results are in general agreement with the current consensus that flatter minima have better generalization (Hinton & van Camp, 1993; Hochreiter & Schmidhuber, 1997; Baldassi et al., 2016; Chaudhari et al., 2016; Zhang et al., 2018b). It was recently suggested that the landscape-dependent noise in SGD-based algorithms can drive the system towards flat minima (Feng & Tu, 2020). However, in the large batch setting, the SSGD noise is too small to be effective. The additional landscape-dependent noise $\Delta^{(2)}$ in DPSGD, which also depends inversely on the flatness of the loss function (see Eq. 4), is thus critical for the system to find flatter minima in the large batch setting.

## 3 EXPERIMENTAL METHODOLOGY

We implemented SSGD and DPSGD using PyTorch, OpenMPI, and NVidia NCCL. We run experiments on a cluster of 8-V100-GPU x86 servers. For CV tasks, we evaluated on CIFAR-10 (50,000 samples, 178MB). For ASR tasks, we evaluate on SWB-300 (300 hours training data, 4,000,000 samples, 30GB) and SWB-2000 (2000 hours training data, 30,000,000 samples, 216GB)[3]. We evaluate on 12 state-of-the-art NN models: 10 CNN models and 2 6-layer bi-directional LSTM models. We summarize the model size and training time in Table 1. We refer readers to Appendix D for software implementation, hardware configuration, dataset and Neural Network (NN) model details.

## 4 EXPERIMENTAL RESULTS

All the large batch experiments are conducted on 16 GPUs (learners) if not stated otherwise. Batches are evenly distributed among learners, e.g., each learner uses a local batch size of 128, when the overall batch size is 2048. A learner randomly picks a neighbor with which to exchange weights in each iteration (Zhang et al., 2020).

### 4.1 SSGD AND DPSGD COMPARISON ON CV TASKS

For CIFAR-10 tasks, we use the hyper-parameter setup proposed in (Liu, 2020): a baseline batch size 128 and learning rate 0.1 for the first 160 epochs, learning rate 0.01 for the next 80 epochs, and learning rate 0.001 for the remaining 80 epochs. Using the same learning rate schedule, we keep increasing the batch size up to 8192. Table 2 records test accuracy under different batch sizes. Model accuracy consistently deteriorates beyond batch size 1024 because the learning rate is too small for the number of parameter updates.

To improve model accuracy beyond batch size 1024, we apply the linear scaling rule (i.e., linearly increase learning rate w.r.t batch size) (He et al., 2015; Zhang et al., 2019a; Goyal et al., 2017; Zhang et al., 2016b;a). We use learning rate 0.1 for batch size 1024, 0.2 for batch size 2048, 0.4 for batch size 4096, and 0.8 for batch size 8192. Table 3 compares SSGD and DPSGD performance running with 16 GPUs (learners). SSGD and DPSGD perform comparably up to batch size 4096. When the batch size increases to 8192, DPSGD outperforms SSGD in all but one case. Most noticeably, SSGD

---

[3]SWB-2000 training is more challenging than ImageNet. It takes over 200 hours on 1 V100 GPU to finish training SWB-2000. SWB-2000 has 32,000 highly unevenly distributed classes whereas ImageNet has 1000 evenly distributed classes.

|  | Batch Size | | | | | | |
|---|---|---|---|---|---|---|---|
|  | 128 | 256 | 512 | 1024 | 2048 | 4096 | 8192 |
| EfficientNet-B0 | 87.51 | 89.32 | 91.28 | **91.92** | 90.62 | 88.00 | 84.85 |
| SENet-18 | **95.18** | 94.84 | 94.83 | 94.52 | 93.83 | 92.94 | 91.69 |
| VGG-19 | 93.51 | **93.78** | 93.35 | 93.12 | 92.64 | 91.82 | 87.76 |
| ResNet-18 | **95.44** | 95.26 | 95.08 | 94.59 | 94.96 | 92.98 | 91.24 |
| DenseNet-121 | 95.06 | 95.27 | **95.42** | 95.11 | 94.81 | 93.09 | 92.34 |
| MobileNet | 89.53 | 90.96 | **92.39** | 92.24 | 91.22 | 89.54 | 86.59 |
| MobileNetV2 | 90.52 | 92.93 | 94.17 | **94.99** | 93.71 | 91.97 | 89.81 |
| ShuffleNet | 90.4 | 92.27 | 92.82 | **93.15** | 91.94 | 90.59 | 87.81 |
| GoogleNet | 94.99 | 95.06 | 94.97 | **95.32** | 94.05 | 92.78 | 91.09 |
| ResNext-29 | 95.35 | **95.66** | 95.31 | 95.42 | 94.24 | 93.00 | 91.06 |

Table 2: CIFAR-10 accuracy (%) with different batch size. Across runs, learning rate is set as 0.1 for first 160 epochs, 0.01 for the next 80 epochs and 0.001 for the last 80 epochs. Model accuracy consistently deteriorates when batch size is over 1024. Bold text in each row represents the highest accuracy achieved for the corresponding model, e.g., EfficientNet-B0 achieves highest accuracy at 91.92% with batch size 1024.

|  |  | Eff-B0 | SE-18 | VGG | Res-18 | Dense-121 | Mobile | MobileV2 | Shuffle | Google | ResNext-29 |
|---|---|---|---|---|---|---|---|---|---|---|---|
| bs=128 lr=0.1 | Baseline | 87.51 | 95.18 | 93.51 | 95.44 | 95.06 | 89.53 | 90.52 | 90.40 | 94.99 | 95.35 |
| bs=1024 lr=0.1 | SSGD | **91.92** | 94.52 | 93.12 | 94.59 | 95.11 | 92.24 | **94.99** | 93.15 | **95.32** | 95.42 |
|  | DPSGD | 91.69 | **94.55** | **93.15** | **94.98** | **95.12** | **92.52** | 94.36 | **93.55** | 95.18 | **95.72** |
| bs=2048 lr=0.2 | SSGD | **91.69** | 94.36 | 92.64 | **94.96** | 95.11 | 91.72 | 94.24 | **92.91** | 94.76 | 94.19 |
|  | DPSGD | 91.06 | **94.70** | **93.05** | 94.86 | **95.32** | **92.72** | **94.51** | 92.89 | **94.80** | **95.30** |
| bs=4096 lr=0.4 | SSGD | **91.62** | 94.28 | 92.68 | 94.30 | 94.72 | 91.68 | **94.25** | **92.67** | 94.36 | 93.21 |
|  | DPSGD | 91.23 | **94.58** | **92.72** | **94.78** | **95.24** | **92.03** | 94.12 | 92.20 | **94.99** | **94.32** |
| bs=8192 lr=0.8 | SSGD | 10 | 10 | 87.11 | 92.70 | 92.79 | 91.10 | **93.22** | 92.09 | 93.72 | 92.38 |
|  | DPSGD | **91.13** | **90.48** | **90.52** | **94.34** | **94.79** | **91.80** | 93.09 | **92.36** | **93.84** | **92.55** |

Table 3: CIFAR-10 comparison for batch size 2048, 4096 and 8192, with learning rate set as 0.2, 0.4 and 0.8 respectively. All experiments are conducted on 16 GPUs (learners), with batch size per GPU 128, 256 and 512 respectively. Bold texts represent the best model accuracy achieved given the specific batch size and learning rate. When batch size is 8192, DPSGD significantly outperforms SSGD. The batch size 128 baseline is presented for reference. bs stands for batch-size, lr stands for learning rate.

diverges in EfficientNet-B0 and SENet-18 when the batch-size is 8192. Figure 6 in Appendix C details model accuracy progression w.r.t epochs in each setting.

To better understand the loss landscape in SSGD and DPSGD training, we visualize the landscape contour 2D projection and Hessian 2D projection, using the same mechanism as in (Li et al., 2018). For both plots, we randomly select two $N$-dim vectors (where $N$ is the number of parameters in each model) and multiply with a scaling factor evenly sampled from -0.1 to 0.1 in a $K \times K$ grid to generate $K^2$ perturbations of the trained model. To produce a contour plot, we calculate the testing data loss of the perturbed model at each point in the $K \times K$ grid. Figure 3 depicts the 2D contour plot for representative models (at the end of the 320th epoch) in a $50 \times 50$ grid. DPSGD training leads not only to a lower loss but also much more widely spaced contours, indicating a flatter loss landscape and more generalizable solution. For the Hessian plot, we first calculate the maximum eigen value $\lambda_{max}$ and minimum eigen value $\lambda_{min}$ of the model's Hessian matrix at each sample point in a 4x4 grid. We then calculate the ratio $r$ between $|\lambda_{min}|$ and $|\lambda_{max}|$. The lower $r$ is, the more likely it is in a convex region and less likely in a saddle region. We then plot the heatmap of this $r$ value in this 4x4 grid. The corresponding models are trained at the 16-th epoch (i.e. the first 5% training phase) and the corresponding Hessian plot Figure 4 indicates DPSGD is much more effective at avoiding early traps (e.g., saddle points) than SSGD.

*Summary* DPSGD outperforms SSGD for 9 out of 10 CV tasks in the large batch setting. Moreover, SSGD diverges on the EfficientNet-B0 and SENet-18 tasks. DPSGD is more effective at avoiding early traps (e.g., saddle points) and reaching better solutions than SSGD in the large batch setting.

### 4.2 SSGD AND DPSGD COMPARISON ON ASR TASKS

For the SWB-300 and SWB-2000 tasks, we follow the same learning rate schedule proposed in (Zhang et al., 2019a): we use learning rate 0.1 for baseline batch size 256, and linearly warmup

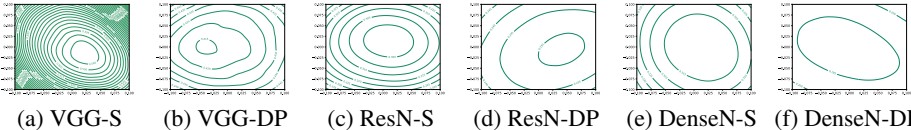

Figure 3: CIFAR-10 2D contour plot. The more widely spaced contours represent a flatter loss landscape and a more generalizable solution. The distance between each contour line is 0.005 across all the plots. We plot against the model trained at the end of 320th epoch. VGG: VGG-19, ResN: ResNet-18, DenseN: DenseNet-121, -S: -SSGD, -DP: -DPSGD

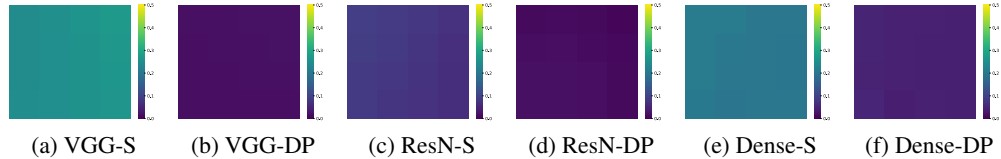

Figure 4: CIFAR-10 Hessian heatmap on a 4x4 grid. The lower value (i.e. a cooler color) indicates the corresponding point is less likely in a saddle. We plotted against the models at the end of the 16th epoch. DPSGD is much more effective at avoiding early traps (e.g., saddle points) than SSGD. VGG: VGG-19, ResN: ResNet-18, DenseN: DenseNet-121, -S: -SSGD, -DP: -DPSGD

learning rate w.r.t the baseline batch size for the first 10 epochs before annealing learning rate by $\frac{1}{\sqrt{2}}$ for the remaining 10 epochs. For example, when using a batch size 2048, we linearly warmup the learning rate to 0.8 by the end of the 10th epoch before annealing. Table 4 illustrates heldout loss for SWB-300 and SWB-2000. In the SWB-300 task, SSGD diverges beyond batch size 2048 and DPSGD converges well until batch size 8192. In the SWB-2000 task, SSGD diverges beyond batch size 4096 and DPSGD converges well until batch size 8192. Figure 7 in Appendix C details heldout loss progression w.r.t epochs.

*Summary* For ASR tasks, SSGD diverges whereas DPSGD converges to baseline model accuracy in the large batch setting.

### 4.3 NOISE-INJECTION AND LEARNING RATE TUNING

In 4 out of 12 studied tasks, a large batch setting leads to a complete divergence in SSGD: EfficientNet-B0, SENet-18, SWB-300 and SWB-2000. As discussed in Section 2, the intrinsic landscape-dependent noise in DPSGD effectively helps escape early traps (e.g., saddle points) and improves training by automatically adjusting learning rate. In this section, we demonstrate these facts by systematically adding Gaussian noise (the same as the $SSGD^*$ algorithm in Section 2) and decreasing the learning rate. We find that SSGD might escape early traps but still results in a much inferior model compared to DPSGD.

**Noise-injection** In Figure 1, we systematically explore Gaussian noise injection with mean 0 and standard deviation (std) ranging from 10 to 0.00001 via binary search (i.e. roughly 20 configurations for each task). We found in the vast majority of the setups, noise-injection cannot escape early traps. In EfficientNet-B0, only when std is set to 0.04, does the model start to converge, but to a very bad loss (test accuracy 22.15% in SSGD vs 91.13% in DPSGD). In SENet-18, when std is set to 0.01, the model converges to a reasonable accuracy (84.86%) but still significantly lags behind its DPSGD counterpart (90.48%). In the SWB-300 case, when std is 0.01, SSGD shows an early sign of

|  | SWB-300 | | | SWB-2000 | | |
|---|---|---|---|---|---|---|
|  | bs2048 | bs4096 | bs8192 | bs2048 | bs4096 | bs8192 |
| SSGD | 1.58 | 10.37 | 10.37 | 1.46 | 1.46 | 10.37 |
| DPSGD | 1.59 | 1.60 | 1.66 | 1.45 | 1.47 | 1.47 |

Table 4: Heldout loss comparison for SSGD and DPSGD, evaluated on SWB-300 and SWB-2000. There are 32000 classes in this task, a held-out loss 10.37 (i.e. $ln^{32000}$) indicates a complete divergence. bs stands for batch size.

|  |  | Eff-B0 | SE-18 | VGG | Res-18 | Dense-121 | Mobile | MobileV2 | Shuffle | Google | ResNext-29 |
|---|---|---|---|---|---|---|---|---|---|---|---|
| lr=0.8 | SSGD | 10.00 | 10.00 | 87.11 | 92.7 | 92.79 | 91.10 | **93.22** | 92.09 | 93.72 | 92.38 |
|  | DPSGD | **91.13** | 90.48 | 90.52 | **94.34** | **94.79** | **91.80** | 93.09 | **92.36** | **93.84** | 92.55 |
| lr=0.4 | SSGD | 88.61 | 92.84 | 91.06 | 91.98 | 93.42 | 91.13 | 93.11 | 91.54 | 92.85 | 89.70 |
|  | DPSGD | 89.80 | **94.00** | **91.93** | 93.91 | 94.32 | 91.38 | 93.14 | 91.68 | 93.49 | **92.79** |
| lr=0.2 | SSGD | 88.03 | 92.41 | 90.51 | 92.13 | 92.98 | 88.38 | 91.68 | 90.14 | 92.44 | 91.31 |
|  | DPSGD | 87.69 | 93.11 | 91.59 | 93.30 | 94.28 | 89.18 | 92.52 | 90.13 | 93.41 | 91.79 |

Table 5: CIFAR-10 with batch size 8192. By reducing learning rate, SSGD can escape early traps but still lags behind DPSGD. Bold text in each column indicates the best accuracy achieved for that model across different learning rate and optimization method configurations. DPSGD consistently delivers the most accurate models.

|  |  | SWB-300 (bs4096) | SWB-300 (bs8192) | SWB-2000 (bs 8192) |
|---|---|---|---|---|
| lr*=0.1 | SSGD | 10.37 | 10.37 | 10.37 |
|  | DPSGD | **1.60** | **1.66** | **1.47** |
| lr=0.05 | SSGD | 10.37 | 10.37 | 10.37 |
|  | DPSGD | 1.65 | 1.73 | 1.48 |
| lr=0.025 | SSGD | 1.76 | 10.37 | 1.51 |
|  | DPSGD | 1.77 | 1.80 | 1.52 |
| lr=0.0125 | SSGD | 1.92 | 2.05 | 1.58 |
|  | DPSGD | 1.94 | 2.00 | 1.59 |

Table 6: Decreasing learning rate for SWB-300 and SWB-2000 (bs stands for batch-size). Bold text in each column indicates the best held-out loss achieved across different learning rate and optimization method configurations for the corresponding batch size. DPSGD consistently delivers the most accurate models. *The learning rate used here corresponds to batch size 256 baseline learning rate, and we still adopt the same learning rate warmup, scaling and annealing schedule. Thus when this learning rate reduces by $x$, the overall effective learning rate also reduces by $x$.

converging for the first 3 epochs before it starts to diverge. In the SWB-2000 case, we didn't find any configuration that can escape early traps. Figure 1 characterizes our best-effort Gaussian noise tuning and its comparison against SSGD and DPSGD. A plausible explanation is that Gaussian noise injection escapes saddle points very slowly, since Gaussian noise is isotropic and the complexity for finding local minima is dimension-dependent (Ge et al., 2015). Deep Neural Networks are usually over-parameterized (i.e., high-dimensional), so it may take a long time to escape local traps. In contrast, the heightened landscape-dependent noise in DPSGD is anisotropic (Chaudhari & Soatto, 2018; Feng & Tu, 2020) and can drive the system to escape in the right directions.

**Learning Rate Tuning** Table 5 and Table 6 compare model quality (measured in either test accuracy or held-out loss) for different learning rates in the large batch size setting, for CV and ASR tasks. By using a smaller learning rate, SSGD can escape early traps, yet it consistently lags behind DPSGD in the large batch setting.

*Summary* By systematically introducing landscape-independent noise and reducing the learning rate, SSGD could escape early traps (e.g., saddle points), but results in much inferior models compared to DPSGD in the large batch setting.

### 4.4 END-TO-END RUN-TIME COMPARISON AND ADVICE FOR PRACTITIONERS

Please refer to Appendix F.

## 5 CONCLUSION

In this paper, we investigate why DPSGD outperforms SSGD in the large batch training. Through detailed analysis on small-scale tasks and an extensive empirical study of a diverse set of modern DL tasks, we conclude that the landscape-dependent noise, which is strengthened in the DPSGD system, brings two benefits in the large batch setting: (1) It adaptively adjusts the effective learning rate according to the loss landscape, helping convergence. (2) It enhances search in weight space to find flat minima with better generalization. Based on our findings, we recommend that DDL practitioners consider DPSGD as an alternative when the batch size must be kept large, e.g., when a shorter run time to reach a reasonable solution is desired.

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
