# OpenReview forum: "Why Does Decentralized Training Outperform Synchronous Training In The Large Batch Setting?"
_ICLR.cc/2021/Conference — Reject_

### Official Review · AnonReviewer2 · 2020-10-26
**Interesting take on gossip-based algorithms for deep learning**

**Rating:** 5
**Confidence:** 4

**Review:**

#### Summary
This paper posits that gossip-based SGD methods for distributed deep learning are more stable, and generalize better, than synchronous SGD in large-batch settings. There is some intuitive discussion, but the hypothesis is mostly validated empirically on MNIST, CIFAR10, and several ASR tasks, for a wide variety of model architectures. I found this hypothesis very interesting, and to be of significant value to the community. I also found the approach sufficiently novel. The empirical evidence was convincing, but was restricted to very small-scale tasks, and so on larger datasets, it is not clear to me what the limiting batch-size is going to be where one observes the behaviour outlined in this work, or whether such a limiting batch-size even exists. The paper was also missing a study on the magnitude of the consensus error $\Delta^{(2)}$ (via the graph topology) on model performance.

Including a study on i) larger datasets and ii) various consensus errors $\Delta^{(2)}$ via the graph topology, would strengthen the paper, and I would be happy to increase my support for this work.

---
#### Originality
Most of the literature in this area has demonstrated that gossip-based methods converge faster (i.e., in wall-clock time), but to possibly higher error solutions. Previous literature has also demonstrated that this gap can be bridged with longer training. By consider extremely large batch-sizes, this work demonstrates that there may be a regime in which gossip-based methods are more stable and generalize better.

The paper first sets out to intuitively describe this phenomenon by decomposing the noise in the gradient-based updates. The decomposition is provided as the sum of a "true" gradient term and an orthogonal component. I quite liked this approach to the decomposition because it then allows the author to describe what they refer to as the effective learning-rate — a quantity that they can empirically measure in practice. While I generally liked this exposition, there is a minor error, which seems inconsequential for the rest of the paper: it is stated that these noise terms $\eta$ are unbiased, which as far as I understand doesn't appear to be correct. Generally speaking, under unbiased sampling, the noise term $\eta$ has zero mean if the gradients are evaluated at $w_\alpha$ (or are linear in their argument), but not if they are evaluated at $w_j$, as in DPSGD; i.e., $E[\frac{1}{n}\sum_{j}g_j(w_j)] \neq E[g(w_a)]$, where the expectation is with respect to the mini-batch sampled across all workers. Though more generally, I also liked the highlighted relationship between the noise strength $\Delta$ and the effective learning learning rate $\alpha_\epsilon$, which does not depend on this (possibly erroneous) zero-mean property.

The hypothesis is thereafter tested through extensive experiments on MNIST, CIFAR10, and various ASR tasks for a broad range of hyperparameters.  Both the goal of the paper, and the approach taken to address it (i.e., through the reasoning of effective learning rates), are sufficiently novel in my opinion.

---
#### Quality and Significance

Large batch deep-learning is of significant interest for parallelizing deep learning workflows. There do exist known limitations for scaling large-batch training, and therefore finding a regime (in the limit of large batches) in which certain methods are not only more efficient, but also more effective, is certainly of value to the community.

I am not familiar with ASR baselines, but the vision baselines look reasonable. I was actually quite surprised by Figure 1. I was not aware of this behaviour for SGD with large batch sizes on CIFAR10. However, I would be interested to see if this observation is particular to CIFAR10 on vision tasks, or whether similar results hold on ImageNet, which is much more stable to large batch-sizes. For example, training on ImageNet with the same batch-size is known to work well. One question I have concerns a subtle point that could explain the results you’re seeing in Fig.1: is the loss (and accuracy) being computed before or after averaging/gossiping the weights? The step-size is scaled according to the batch-size, and your large-batch update is only complete after averaging/gossiping, therefore $w_j$ is a poorly specified set of model weights, and the curves should be evaluated at $w_a$.

More generally, modulo this important distinction in my opinion, I found the empirical evidence convincing. However, my main concern is that the analysis was only on very small-scale tasks, and so on larger datasets, it is not clear to me what the limiting batch-size is going to be where one observes the behaviour outlined in this work, or whether such a limiting batch-size even exists.

What was missing was also a study on the effect of the graph topology. It is stated in the appendix that 16 learners with randomized communication was found to work well, but I think crucial to this discussion is how the consensus error $\Delta^{(2)}$, which depends on the graph topology, affects this aforementioned behaviour.

Are the observations of the 2D contour and hessian plots consistent across multiple sampled directions? I ask because the parameters are 100-dimensional, and so it may not be the case that the level sets are entirely characterized by these 2 random vectors.

---
#### Clarity

The work is clear for the most part. Some minor comments on the exposition
* The last approximation in equation 4 is not clear to me (e.g., what is the double subscript on the weights), though I agree with the following sentence, that the consensus error is going to be larger in sharper landscapes.
* Not sure what qualifies as a “generic algorithm” so I would remove the phrase "To the best of our knowledge, we are unaware of any generic algorithm that can improve SSGD large batch training on this many models/tasks."
* I would temper sharp vs flat minima arguments, since these arguments have been shown to depend on the parameterization of the objective.
* Decentralized training can be synchronous (e.g., via all-reduce), and so I don’t think the real distinction (i.e., gossip-based vs globally synchronous methods) is well captured in the title or the exposition.
* The common argument that large batches converge to sharp minima is still a largely experimental idea, and so basing an argument on these principles does not provide convincing evidence (especially since such arguments depend on the parameterization of the objective).
* How is the complexity of escaping a saddle point computed, and why is it assumed that the models in Figure 1 are stuck in saddle points. Phrases like this should be somewhat tempered, or at least discussed in more detail: “However, this is not a good solution for high-dimensional DL training as shown in the blue curves of Figure 1. One possible reason is that the complexity of escaping a saddle point by adding isotropic noise has a polynomial dependency on the dimension of the parameter space, so adding such noise in a high dimensional space (such as deep learning) does not bring significant benefits"

---

> ### Author Response · Authors · 2020-11-17
> **Response to AnonReviewer2**
>
> Q1. One question I have concerns a subtle point that could explain the results you’re seeing in Fig.1: is the loss (and accuracy) being computed before or after averaging/gossiping the weights? The step-size is scaled according to the batch-size, and your large-batch update is only complete after averaging/gossiping, therefore is a poorly specified set of model weights, and the curves should be evaluated at.
>
> We evaluate loss (and accuracy) at the end of each *epoch* in Figure 1, and similarly in the training progression plots in Appendix C. There is no significant difference between measuring before and after the model averaging as the effect is only one mini-batch update away.
>
> Q2. However, my main concern is that the analysis was only on very small-scale tasks, and so on larger datasets, it is not clear to me what the limiting batch-size is going to be where one observes the behaviour outlined in this work, or whether such a limiting batch-size even exists.
>
> We emphasize that SWB-2000 is a larger scale task than ImageNet: the 2,000 hours of audio in this task corresponds to 30,000,000 training samples (216 GB of storage) that are classified into 32,000 classes that follow a long-tailed, Zipfian distribution. In contrast, ImageNet has 1,200,000 training samples that are classified into 1,000 classes which are more evenly distributed. We mentioned the scale of SWB-2000 in Section 3.
>
> Q3. What was missing was also a study on the effect of the graph topology. It is stated in the appendix that 16 learners with randomized communication was found to work well, but I think crucial to this discussion is how the consensus error , which depends on the graph topology, affects this aforementioned behaviour.
>
> (Zhang et al., 2020) provides a theoretical and empirical analysis of convergence as a function of different graph topologies in DPSGD.  It also demonstrates that when the number of learners is smaller than 16 (as is the case in this submission), there is not much difference between using a randomized graph topology and the fixed graph topology in (Zhang et al. 2020)
>
> Q4.  Are the observations of the 2D contour and hessian plots consistent across multiple sampled directions? I ask because the parameters are 100-dimensional, and so it may not be the case that the level sets are entirely characterized by these 2 random vectors.
>
> For each plot, we randomly sample two vectors and the plotting procedure followed the same practice as in (Li et al., 2018) ). The observations of the 2D contour plot and hessian plots are consistent across multiple randomly sampled pairs of directions. For the MNIST problem, we have diagonalized the Hessian matrix and found that the eigenvalues at the solutions found by DPSGD are smaller than those at solutions found by SSGD, which means the loss landscape near the DPSGD solutions is flatter.
>
> Q5. The last approximation in equation 4 is not clear to me (e.g., what is the double subscript on the weights), though I agree with the following sentence, that the consensus error is going to be larger in sharper landscapes.
>
> In ${\Delta}w_{j,l}$, the first subscript j labels the learner (j=1,2,..n) and the second index labels the component of the weight vector.
>
> Q6. Not sure what qualifies as a “generic algorithm” so I would remove the phrase "To the best of our knowledge, we are unaware of any generic algorithm that can improve SSGD large batch training on this many models/tasks."
>
> Thanks, we will revise
>
> Q7. I would temper sharp vs flat minima arguments, since these arguments have been shown to depend on the parameterization of the objective.
>
> Thanks, we will revise.
>
> Q8. Decentralized training can be synchronous (e.g., via all-reduce), and so I don’t think the real distinction (i.e., gossip-based vs globally synchronous methods) is well captured in the title or the exposition.
>
> Thanks, we will consider revising the paper title.
>
> Q9. The common argument that large batches converge to sharp minima is still a largely experimental idea, and so basing an argument on these principles does not provide convincing evidence (especially since such arguments depend on the parameterization of the objective).
>
> Thanks, we will revise.
>
>
> Q10. How is the complexity of escaping a saddle point computed, and why is it assumed that the models in Figure 1 are stuck in saddle points.
>
> In the experiments illustrated in Figure 1,  we suspect training is stuck in a saddle point when model quality (measured by test accuracy or heldout loss) did not improve at all or significantly lagged behind the baseline accuracy over the predefined training epochs.

---

> ### Comment · AnonReviewer2 · 2020-11-24
> **Thank you for your responses**
>
> Dear authors,
>
> Thank you for responding to many of my questions. I want to re-emphasize that I think this paper provides an interesting take on gossip algorithms for deep learning.
>
> I can appreciate the response highlighting the size of the ASR task, but the unfortunately, in the context of the vision tasks, the empirical evidence is not sufficiently convincing to support the main argument of the paper, hence my recommendation for a larger vision dataset than CIFAR10. This point also appeared to be raised by some of the other reviewers. The main arguments in this paper crucially depend on the notion of large mini-batches, but what constitutes a large mini-batch depends on the choice of data set. For example, as I indicated in my review, the large batch-size used to provide the arguments on CIFAR10 would actually work perfectly fine on ImageNet, and does not support the main arguments in this paper.
>
> I also understand the logic that graph topology does not make much of a difference for a small number of learners, but the graph topology impacts the diffusion rate, and directly affects the convergence (theoretical constants and empirical rates). The choice of graph topology has been shown to make a large difference for deep learning applications, especially in the large-batch setting, which is the main setting considered in this paper (e.g., Lian et al., NeurIPS 2017; Lian et al., ICML 2018, Assran et al., ICML 2019). Given that the main argument is that the consensus noise can directly improve generalization in the large-batch setting, and the consensus noise is a direct function of the graph topology, I think an examination on the choice of graph topology would've been a valuable addition.
>
> In short, neither of my two main concerns were addressed, but I will keep an open mind when discussing with the other reviewers.

---

### Official Review · AnonReviewer1 · 2020-10-27
**Interesting idea, but lack of solid theoretical supports and in-depth discussion**

**Rating:** 3
**Confidence:** 5

**Review:**

This paper claims that decentralized parallel SGD (DPSGD) performs better than synchronous SGD (SSGD) and noisy version of synchronous SGD (SSGD*) in large batch setting. Theoretically, it shows that the noise in DPSGD is landscape-dependent, which may help generalization. Experimental results on CV and ASR tasks show that DPSGD can outperform baselines when batch size is very large. Meanwhile, DPSGD is observed to adaptively adjust the effective learning rate and converge to flatter minima.

Although the idea in the paper is insightful, most of the claims are lack of sufficient supports and in-depth discussion. For the theoretical part, Eq(4) only shows the differences of noise between SSGD and DPSGD. The explicit relation between Delta^(2) and generalization is not provided. After reading Eq.(4), I still do not know why Delta^(2) can help generalization. In my mind, Delta_S is also landscape-dependent because the covariance matrix of \Delta_S aligns well with Hessian in some cases [1][2].

The explicit relation between decreasing effective learning rate and convergence rate of an optimization algorithm is not provided too. The convergence not only depends on the tendency of the learning rate but its decreasing rate. For an extreme instance, if the effective learning rate decreases rapidly, it is hard to imagine that it will help convergence.

Besides, there are many writing issues in the paper. Please see the details comments below.

Detailed comments:

(1) For a scientific paper, every claim should be objective. There are many subjective claims in this paper. For example, "Recently, ASGD has lost popularity due to....."(in Intro), "One possible reason is that the complexity of escaping a saddle point....."(in Intro), "The poor performance is likely due to......"(in Sec2.1), etc. These claims are lack of sufficient supports  and are not convincing.

(2) There are redundant notations which make the paper hard to follow. For example, the "SSGD+noise" in Figure 1 and "SSGD*" in Figure 2; the subscript of a letter sometimes denotes the learner, sometimes denotes the average or subset average.

(3) Some claims are ambiguous. For example, "It is clear that \Delta^(2)" depends on the loss landscape-it is larger in rough landscapes and smaller in flat landscapes". What is the measure of the loss landscape and the flatness? Both landscape and flatness are descriptive terms and their measures should be clearly introduced. Why does the noise in SSGD not depend on landscape?

(4) The term "Multiple learners" is misleading, especially for SSGD. It is better to name them local workers because they only calculated gradients and SSGD outputs a unified model.

(5) I suggest to use the original format because this version looks too crowded. For example, the space between sections is too small and the paragraphs in the introduction are merged.

In summary, the current version is not ready to be published.

[1] Wen, et al., An Empirical Study of Large-Batch Stochastic Gradient Descent with Structured Covariance Noise

[2] Zhu, et al,. The Anisotropic Noise in Stochastic Gradient Descent: Its Behavior of Escaping from Minima and Regularization Effects.

---

> ### Author Response · Authors · 2020-11-17
> **Response to AnonReviewer1**
>
> Q1. For the theoretical part, Eq(4) only shows the differences of noise between SSGD and DPSGD. The explicit relation between Delta^(2) and generalization is not provided. After reading Eq.(4), I still do not know why Delta^(2) can help generalization. In my mind, Delta_S is also landscape-dependent because the covariance matrix of \Delta_S aligns well with Hessian in some cases [1][2].
>
> The reviewer is correct to point out that the noise $\Delta_{S}$ in SSGD is also landscape dependent. In fact, all SGD-based algorithms have this intrinsic landscape-dependent noise. However, as we stated in our paper (2nd to last paragraph on page3), in SSGD with n synchronized learners,   is proportional to 1/(nB) and becomes too small to be effective in the large batch setting. Therefore, the additional noise term $\Delta^{(2)}$, which also depends on the landscape (Eq.4), is needed to give DPSGD a large enough landscape-dependent noise. We don’t have an exact relation between noise $\Delta^{(2)}$ and generalization. However, from Eq.4, we see that the noise $\Delta^{(2)}$ depends on the hessian of the landscape, i.e., the noise $\Delta^{(2)}$ is larger at a sharp minimum than at a flatter minimum. As we know from stochastic dynamics, a stochastic system will be more likely to stay at a minimum with a smaller noise (lower temperature). Therefore, the landscape-dependent noise $\Delta^{(2)}$ in DPSGD is likely to be responsible for DPSGD to find and stay at a flatter minimum, which is generally believed to have better generalization performance than a sharp minimum.
>
> Q2. The explicit relation between decreasing effective learning rate and convergence rate of an optimization algorithm is not provided too. The convergence not only depends on the tendency of the learning rate but its decreasing rate. For an extreme instance, if the effective learning rate decreases rapidly, it is hard to imagine that it will help convergence.
>
> The effective learning rate depends on the landscape roughness. As shown in Fig. 2a (green line in lower panel), the effective learning rate for DPSGD is reduced only at the beginning of the training process when the loss-landscape is rough (green line in Fig. 2a lower panel) so that DPSGD can avoid overshoot. Once the landscape becomes smoother, $\alpha_{e}$ recovers to be close to its original value $\alpha$, therefore the overall convergence rate is still fast. With this landscape-dependence effective learning rate, DPSGD gets learning rate tunning for free. When the original learning rate $\alpha$ is high, this self-adjusted learning rate tunning allows DPSGD to converge while SSGD cannot converge at all (see Fig. 2a) – note that this self-tunning of the effective learning rate is much weaker in SSGD due to the weak SSGD noise in large batch setting.
>
> Q3. For a scientific paper, every claim should be objective. There are many subjective claims in this paper. For example, "Recently, ASGD has lost popularity due to....."(in Intro), "One possible reason is that the complexity of escaping a saddle point....."(in Intro), "The poor performance is likely due to......"(in Sec2.1), etc. These claims are lack of sufficient supports and are not convincing.
>
> In the revision, we will include more citations and more detailed explanations.
>
> Q4. There are redundant notations which make the paper hard to follow. For example, the "SSGD+noise" in Figure 1 and "SSGD*" in Figure 2; the subscript of a letter sometimes denotes the learner, sometimes denotes the average or subset average.
>
> Thanks! We will improve the notation in the revision.
>
> Q5. Some claims are ambiguous. For example, "It is clear that \Delta^(2)" depends on the loss landscape-it is larger in rough landscapes and smaller in flat landscapes". What is the measure of the loss landscape and the flatness? Both landscape and flatness are descriptive terms and their measures should be clearly introduced. Why does the noise in SSGD not depend on landscape?
>
> By landscape, we mean the variation of the loss function in weight space, i.e., the shape of $L(\vec{w})$ . The flatness of the landscape in different directions of the weight space can be defined by the inverse of the hessian matrix (H in Eq.4) of the loss function at its minimum. The noise in SSGD does depend on landscape, but its amplitude is too small in the large batch setting, rendering it ineffective (see the second to last paragraph on page 3 in our paper for discussion).
>
> Q6. The term "Multiple learners" is misleading, especially for SSGD. It is better to name them local workers because they only calculated gradients and SSGD outputs a unified model.
>
> By “multiple learners”, we mean that SSGD is computed in a distributed manner by multiple learners instead of a single learner.
>
> Q7.  I suggest to use the original format because this version looks too crowded.
>
> Thanks! We will address this in the revision.

---

### Official Review · AnonReviewer3 · 2020-10-29
**This paper gives some empirical comparison of centralized sgd and decentralized sgd for relatively large batch size**

**Rating:** 3
**Confidence:** 4

**Review:**

This paper gives some empirical comparison of centralized sgd and decentralized sgd for relatively large batch size. The authors claim that decentralized sgd performs better than centralized version and that this is because the noise introduced by the decentralized version helps escape the local minima.

While it is claimed that "We show, both theoretically and empirically, that the intrinsic noise in DPSGD can...", it is hardly possible to find any theoretical insight. The theoretical analysis part, Section 2, contains a figure (Figure 2) with an entire subsection (Section 2.2) discussing it, but include no theorem, lemma, or even proposition! There is clearly stated noise comparison between centralized sgd and decentralized sgd and thus it is not clear why one can claim decentralized sgd does better becomes of noise.

Second, as a paper studying the large batch setting, it is necessary to define what large batch means and how batch size relates to the convergence, none of which is included in the paper. In fact, in the theoretical analysis part (Section 2), batch size does not appear at all. Why is large batch size important, or is decentralized sgd always better than centralized one? The entire Section 2 is far away from understandable.

The experiment section also fails to justify the claim. The authors compared decentralized and centralized sgd for batch size = 1024, 2048, 4096, 8192. Are those all large batch, or is 1024 small and the other large? From Table 3 I guess it is the former case. But then a nature question is: does decentralized version perform poorly for small batch size? The experiment section  fails to show a "phase transition", which cannot support the claim. Furthermore, for vision tasks, evaluation on a single dataset (CIFAR10) is certainly not enough. Same thing for ASR task. Most importantly, there is no empirical evidence supporting the claim that decentralized sgd " 1) [it] automatically adjusts the learning rate to improve convergence; 2) [it] enhances weight space search by escaping local traps (e.g., saddle points) to find flat minima with better generalization".  Figure 3 and Figure 4 simply show properties of the neural network and the datasets and has nothing to do with optimization methods. Table 2,3,5,6 only shows final accuracy produced by the optimizers, but say nothing about how the optimizers reach the solutions.

The paper writing is problematic too. A single section (Section 3) talking about methodology with less than 1/3 page is awkward. Discussing empirical results in the analysis part (Section 2) is misleading. It is also not acceptable to have "Please refer to Appendix F." for a whole subsection (4.4). Many sentences do not make sense. For example,  "In a large batch setting, the learning rate must be increased to compensate for the reduced number of parameter updates". Why is number of parameters updates reduced? "While there was anecdotal evidence that DPSGD outperforms SSGD in the large-batch setting". What is the evidence? Which paper claims this? There are numerous typos too.

In short, this paper is sloppy and far away from publishable in its current stage.

---

> ### Author Response · Authors · 2020-11-17
> **Response to AnonReviewer3 (1/2)**
>
> Q1 Clarify what do we mean by theory.
>
> The reviewer’s complaints about lack of theorem, lemma, etc. in our theoretical analysis seems to be based on a narrow notion of what a theory should be, with which we strongly disagree. In general, a theory is an idea or a set of guiding principles to explain something. For example,  even though Einstein’s theory of relativity does not involve “theorem, lemma, or even proposition”, it explains how objects move in space and time, which is verified by empirical observations.  No one can argue the theory of relativity is not a theory due to lack of theorem, lemma, etc. Our theoretical work in this paper follows the same logic of how theoretical work should be done. In particular, our goal is to explain why DPSGD outperforms SSGD in the large batch size setting. By detailed noise analysis in DPSGD and SSGD (Sec. 2), we gain the theoretical idea (insight) that the additional landscape-dependent noise $\Delta^{(2)}$  in DPSGD is responsible for its better performance (convergence and generalization) of DPSGD in the large batch setting. This theoretical insight is then verified/supported by empirical experiments including those reported in Fig. 2.
>
> Q2 Batch size definition
>
> The theoretical analysis in section 2 was done in the large batch size setting for the MNIST dataset with batch size nB=2000 as clearly stated in our paper (last line on page 3).
>
> Q3 Why is large batch size important, or is decentralized sgd always better than centralized one?
>
> Large batch size is important for distributed deep learning. Only when the batch size is sufficiently large is there enough available parallelism for distributed deep learning to accelerate training.
>
> Q4 Furthermore, for vision tasks, evaluation on a single dataset (CIFAR10) is certainly not enough. Same thing for ASR task.
>
> We disagree. We experimented with two dramatically different application domains (speech and vision) using very different neural network models (recurrent and convolutional). We also emphasize that SWB-2000 is a larger scale task than ImageNet: the 2,000 hours of audio in this task corresponds to 30,000,000 training samples (216 GB of storage) that are classified into 32,000 classes that follow a long-tailed, Zipfian distribution. In contrast, ImageNet has 1,200,000 training samples that are classified into 1,000 classes which are more evenly distributed. We mentioned the scale of SWB-2000 in Section 3. In addition, SWB-300 and SWB-2000 are well-established large-scale datasets used in the ASR community for many years, and methods that work well on these benchmark tasks have proven to work well on other ASR tasks such as LibriSpeech or broadcast news transcription.
>
> Q5, But then a natural question is: does decentralized version perform poorly for small batch size?
>
> When the batch size is small, there is not enough parallelism for distributed deep learning to accelerate training and there is no reason for practitioners to choose either SYNC or DPSGD over single learner training.
>
> Q6 Most importantly, there is no empirical evidence supporting the claim that decentralized sgd " 1) [it] automatically adjusts the learning rate to improve convergence; 2) [it] enhances weight space search by escaping local traps (e.g., saddle points) to find flat minima with better generalization".
>
> The empirical evidence supporting these two claims are presented in Fig. 2 and explained in detail in Sec. 2. In Sec. 2.1, the effective learning rate for DPSGD is shown to be reduced at the beginning of the training process when the loss-landscape is rough (green line in Fig. 2a lower panel) so that the DPSGD can converge (green line in Fig.2a upper panel); In Sec. 2.2, the noise strength is shown to be larger in DPSGD (green line in Fig.2b lower panel), which allows it to explore a larger weight space to find flatter minima (see the two contour plots in Fig. 2b).
>
> Furthermore,  we demonstrate that DPSGD (1) enhances weight space search by escaping local traps in Figure 4, which shows DPSGD reaches loss-landscape which is more “convex”.  In Figure 4, we first calculate the maximum eigenvalue $\lambda_{\text{max}}$ and minimum eigenvalue $\lambda_{\text{min}}$ of the model's Hessian matrix at each sample point in a 4x4 grid. We then calculate the ratio $r$ between $|\lambda_{\text{min}}|$ and $|\lambda_{\text{max}}|$. The lower $r$ is, the more likely it is in a convex region and less likely in a saddle region. (2) finds flat minima with better generalization in Figure 3. In Figure 3, the more widely spaced contours represent a flatter loss landscape and a more generalizable solution.

---

> > ### Author Response · Authors · 2020-11-17
> > **Response to AnonReviewer3 (2/2)**
> >
> > Q7  “Figure 3 and Figure 4 simply show properties of the neural network and the datasets and has nothing to do with optimization methods”
> >
> > Figure 3 and 4 demonstrate that DPSGD finds flatter optima and ends up in a more convex region than SSGD, for the *same model and same hyper-parameter setup* (i.e., batch size and learning rate).
> >
> > Q8 The experiment section fails to show a "phase transition", which cannot support the claim. Table 2,3,5,6 only shows the final accuracy produced by the optimizers, but say nothing about how the optimizers reach the solutions.
> >
> > As mentioned in Section 4.1 and 4.2, Figure 6 and Figure 7 in Appendix C record the training progression (test accuracy / held-out loss w.r.t training epochs).
> >
> > Q9  In a large batch setting, the learning rate must be increased to compensate for the reduced number of parameter updates". Why is number of parameters updates reduced?
> >
> > Because the computational cost is fixed (i.e., number of epochs is fixed). When the batch size is larger, the number of updates per epoch is reduced; therefore, the total number of parameter updates is reduced.
> >
> > Q10  “While there was anecdotal evidence that DPSGD outperforms SSGD in the large-batch setting". What is the evidence? Which paper claims this?
> >
> > The quoted sentence the reviewer refers to appears in the abstract, and it is customary not to cite papers in the abstract. In the Introduction, we wrote “However, it is observed that SSGD with large batch size leads to large training loss and inferior model quality for ASR tasks (Zhang et al., 2019b), as illustrated in Figure 1a (red curve).”

---

### Official Review · AnonReviewer4 · 2020-10-29
**Comprehensive experimental study. Paper formatting should be improved.**

**Rating:** 6
**Confidence:** 3

**Review:**

The paper did comprehensive experimental study on how DPSGD and SSGD converge in different tasks. Some intuition is provided to explain why DPSGD usually performs better than SSGD under large batch settings. Training with large batch size is a very important topic nowadays to speedup deep learning pipelines. In this paper different models, learning rates, and datasets are evaluated. Overall I feel the experiments show a good sign that decentralization is helpful in this case.

In my opinion the experiments in this paper are valuable, but I would be more happy to see this paper to inspire some solid theoretical studies on exactly why decentralized (and other tricks) can help the large batch training problem (and when they will fail).

Some additional things I would love to see in the revision:
* comparing with SSGD with "warmup" where the learning rate increases from a small value at the beginning of training. This kind of warmup lr schedule has been shown effective in some existing work (for example, Goyal, Priya, et al. "Accurate, large minibatch sgd: Training imagenet in 1 hour." arXiv preprint arXiv:1706.02677 (2017).), and should mitigate that convergence issue mentioned in this paper.
* code to help other practitioners verify the results

I wanted to give a accept but the formatting of this paper has a lot of room to improve. To list a few:
* duplicated references in the reference section
* page 2 footnote 2, the line below equation (2): quote should be fixed
* sections titles formatting (capital letter or not) should be consistent

---

> ### Author Response · Authors · 2020-11-17
> **Response to AnonReviewer4**
>
> Q1. Learning rate warmup.
>
> (1) For the ASR tasks we evaluated, we used learning rate warmup techniques (please refer to the first 4 lines in Section 4.2 for the exact learning rate scheme that we experimented with).
> (2) For the CV tasks, we also experimented with various smaller learning rates for all the studied models in the batch-size=8192 setting. (Table 5)
> (3) It worth noting that for all the SSGD/DPSGD comparisons, we use the same learning rate scheme and DPSGD consistently outperforms SSGD in the large batch setting. As analyzed in the paper, DPSGD automatically adjusts the learning rate (smaller at the beginning and then gradually increasing) due to the loss-landscape dependent system noise. In other words, DPSGD performs learning rate warmup for free, without the user’s needing to select a warmup schedule.
>
> Q2. Code availability
>
> We plan to open-source our code in the future.
>
> Q3. Various formatting issues
>
> We appreciate the reviewer’s feedback. We will address reviewers’ concerns in our revision.

---

### Decision · Program_Chairs · 2021-01-07
**Final Decision**

**Decision:**

Reject

**Comment:**

This paper did experimental studies on how DPSGD and SSGD converge in different tasks. Some concerns were raised regarding the clarity, some unjustified claims, baselines and etc, and partially addressed after the rebuttal and discussions. However, some critical concerns remains. The reviewers agreed that the paper would be more appealing if these concerns can be well addressed.